# The Effect of Placement and Group Size on the Use of an Automated Brush by Groups of Lactating Dairy Cattle

**DOI:** 10.3390/ani13040760

**Published:** 2023-02-20

**Authors:** Borbala Foris, Negar Sadrzadeh, Joseph Krahn, Daniel M. Weary, Marina A. G. von Keyserlingk

**Affiliations:** Animal Welfare Program, Faculty of Land and Food Systems, The University of British Columbia, 2357 Main Mall, Vancouver, BC V6T 1Z6, Canada

**Keywords:** welfare, sickness behavior, enrichment

## Abstract

**Simple Summary:**

Indoor housed dairy cows are highly motivated to scratch themselves using mechanical brushes. Many farms provide brushes to cows, yet no commercial brushes to date capture how the brushes are used by the cows in the pen. We developed an automated brush and tested how much cows use it at four different group sizes (60, 48, 36, and 24 cows) and with different brush locations in the pen. We found that cows used the brush for longer when it was close to the feed and water and when they were housed in smaller groups. We suggest that future studies provide groups with multiple brushes to better understand the influence on brushing behavior of cows.

**Abstract:**

Mechanical brushes are often provided on dairy farms to facilitate grooming. However, current brush designs do not provide data on their use, and thus little is known about the effects of group size and placement of brushes within the pen. The objectives of this study were to automatically detect brush use in cow groups and to investigate the influence of (1) group size and the corresponding cow-to-brush ratio and (2) brush placement in relation to the lying stalls and the feeding and drinking areas. We measured brush use in groups of 60, 48, 36, and 24 cows, with the brush placed either in the alley adjacent to the feed bunk and water trough or in the back alley. Cows used the brush for longer when it was placed in the feed/water alley compared to when placed in the back alley. Average brush use per cow increased when cows were housed in smaller groups, but the brush was never in use more than 50% of the day, regardless of group size. We conclude that brush use increases when availability is increased and when the brush is placed closer to the feed and water.

## 1. Introduction

Mechanical brushes are provided on dairy farms to facilitate grooming behavior, helping cows to remove dirt and parasites [1,2]. Increased grooming improves skin health and lowers the risk of disease [3]. In contrast, a reduction in grooming is one of a coordinated suite of sickness behaviors [4], and thus may be a sensitive indicator of health problems. Having access to a brush appears to be an important resource for cattle, as cows are willing to push as much weight to access a brush as to access fresh feed [5]. Grooming has also been proposed as a potential positive welfare marker, indicating that cows provided with access to a mechanical brush are able to have enjoyable experiences [6].

Despite the growing interest from farmers to install mechanical brushes on their farms, and their recognized potential for providing welfare-relevant information [7], no commercial brush to date collects and provides any data on its use. Accordingly, little is known about the internal and external factors that influence the brush use of dairy cows. Social status has been found to influence brush use in healthy dry cows, with dominant animals using the brush longer than subordinates, independent of agonistic interactions at the brush [8]. Group size and the placement of the brush could potentially modulate brush use, and automated data collection about these influences could facilitate the integration of brushing behavior into on-farm data streams to improve animal welfare. However, there are no validated guidelines about where or how many brushes should be placed to promote brush use. To our knowledge, the only guidance is an industry recommendation that suggests one brush per 60 cows (e.g., Black and Bewley, n.d.), however, this number appears to be based on practical experience rather than peer-reviewed research. With regards to brush placement within the pen, one previous study found that the probability and frequency of brush use events was higher when the brush was placed closer to the feed [9], although to our knowledge no study has observed how the duration of brush use varies by brush location. 

In summary, the research problem is twofold: there is a lack of scientific knowledge about how cows use mechanical brushes in groups of different sizes and with varying locations, and there are no practical devices available to automatically monitor brushing behavior in dairy cow groups. To address this research gap, we aimed to establish an automated brush prototype and to investigate how the use of a single mechanical brush is affected by (1) increasing group sizes and (2) placement of the brush in relation to proximity to feed and water. We hypothesized that cows would use the brush more when housed in smaller groups and when the brush was placed closer to the feed and water.

## 2. Materials and Methods

We measured the brush use of pregnant lactating Holstein cows (mean ± SD, parity: 2.4 ± 1.4; DIM: 255.6 ± 56.2) housed at The University of British Columbia (UBC) Dairy Education and Research Centre in Agassiz, BC, Canada, between September 2021 and January 2022. Cows were housed in a free stall barn with lying stalls bedded with sand; the cow-to-stall ratio was 1:1. Fresh Total Mixed Ration (9% alfalfa hay, 39% corn silage, 28% grass silage, 24% concentrate and mineral mix) was delivered twice daily at approximately 0600 h and 1530 h; cows accessed feed from a post-and-rail feed barrier providing 0.6 m per cow [10]. Cows had *ad libitum* access to water and were removed from the pen twice daily for milking at approximately 0730 h and 1730 h. 

One cylindrical-shaped mechanical brush (Luna, Lely, Maassluis, The Netherlands) was placed in a pen designed to house 60 cows and adapted to this study by the inclusion of an additional alley (Figure 1). The brush was mounted 0.4 m above the shoulder height of the shortest cow according to manufacturer instructions. At both locations tested the brush was placed such that the rotation axis was transverse to the movement direction of manure scrapers, allowing cows to step over the moving scraper while brushing. The brush was fitted with an electronic interface and two light sensors to automatically detect when the swinging brush arm is deflected from the vertical position and the brush rotates. Brush rotation was 30 rpm in the direction opposite to that of the push, and stopping if the brush remained in the vertical position for 4 s. Rotation direction, start times and durations were recorded continuously on a MicroSD card in the electronic interface attached to the brush, and saved daily on a computer while the cows were away from the pen during milking. In a preliminary validation, we found that brush use events in the data (i.e., a new event corresponding to each change in rotation direction and rotation re-start) did not necessarily reflect separate brush use bouts by the cows. Preliminary observations indicated that cows often stop and resume brushing, or multiple cows take turns in using the brush, indicating that brush rotation frequency may be influenced by the general barn environment and its daily variation (e.g., animal and human movement). However, the total daily brush use duration by the group of cows was well represented by the automatically collected brush rotation data. Therefore, instead of brush use frequency, we calculated total daily rotation time to reflect the daily brush use duration by the group.

Cows were tested in two replicates; in each replicate we used a convenience sample of 60 unique pregnant cows. The size of the pen was modified to maintain constant space per cow (9.8 m^2^/cow) and stocking densities of cows to lying stalls and feeding areas for groups of 60, 48, 36, and 24 cows. We measured brush use when the brush was placed at two different locations: the feed alley adjacent to the feed and water, or the alley furthest away from the feed and water (i.e., the back alley). Cows exited and entered the pen for milking from the back alley (Figure 1). 

Cows in each group of 60 were allowed to habituate to the pen for one week to ensure that all cows have comparable experiences with the brush. The order of brush location treatments was reversed between replicates. For each brush location, we first tested the complete group of 60 cows and then varied the group size between 48, 36, and 24 cows quasi-randomly, ensuring different group size orders between locations and replicates. Group size change was implemented by including or excluding random cows from the treatment group in order to minimize social stress associated with regrouping [11]. Data from days when the group size changed were excluded from analysis. Daily temperature data was obtained from the “Agassiz RCS” weather station and downloaded from the Government of Canada website. 

We excluded days with missing data due to technical errors with writing to the microSD card in the brush interface and analyzed group-level brush use data for a minimum of two days (mean ± SD: 3.7 ± 1.9 days) for each group size and brush placement treatment using R 4.2.0 (R Core Team, Vienna, Austria). For each treatment and replicate (four group sizes, two locations, two replicates; n = 16), we calculated the average daily brush use per cow across days and used this as a dependent variable in a linear model including brush location (two levels), group size, and replicate (two levels) as fixed effects. We did not detect any significant interactions so these terms were excluded from the final model. We also investigated the difference in the average daily temperature between replicates using a separate linear model. To understand brush use levels during different times of the day, we calculated the average hourly brush use during daytime (0600 h to 1800 h) and nighttime (1800 h to 0600 h) and investigated their difference using a linear mixed model with time of day (two levels: daytime, nighttime) as a fixed effect and group (defined by the combination of group size, brush location, and replicate) as a random effect. Model residuals were plotted and inspected visually to confirm homoscedasticity and normality.

## 3. Results

Group-level brush use ranged between 2.6 and 10.1 h/day. The mean ± SD of daily brush use for different group sizes was: 24 cows, 3.8 ± 0.6 h; 36 cows, 5.5 ± 1.3 h; 48 cows, 5.6 ± 1.4 h; 60 cows, 7.6 ± 1.8 h. Across treatments, brush use averaged 8.5 ± 2.0 min/day/cow (mean ± SD; Figure 2).

Average daily brush use per cow decreased with increasing group size (Figure 2, by 0.5 ± 0.2 min/day/cow for each additional group of 12 cows added; F_1,12_ = 5.5, *p* = 0.04). Cows used the brush more when it was placed in the feed alley (an increase of 1.4 ± 0.5 min/day/cow, F_1,12_ = 7.06, *p* = 0.02) and brush use was lower during the second replicate (by 2.1 ± 0.5 min/day/cow, F_1,12_ = 16.33, *p* = 0.002). The average daily temperature was also lower during the second replicate (by 8.9 ± 2.5 °C, *p* = 0.003). The average hourly brush use in the groups did not differ between daytime and nighttime.

## 4. Discussion

Average brush use in the current study was slightly higher than values reported in previous work on lactating dairy cows (2–3 min/cow/d in [12]; 5–7 min/cow/d in [13]); this difference may be explained in part by differences in group size, cow-to-brush ratio, and brush location. For instance, Mandel et al. [12] grouped 70–80 cows with access to two brushes that were 3 m and 16 m from the feed bunk. DeVries et al. [13] grouped six groups of 12 cows with access to one brush that was ~4 m from the feed bunk. Stage of lactation may also be important given that previous work on non-lactating cows found much higher brush use (31.5 ± 17.7 min/cow/d by [14], and 27.4 ± 21 min/cow/d by [8]). Differences in time budget between cows in different stages of lactation may warrant investigation.

Cows that were housed in smaller group sizes (with a corresponding lower cow-to-brush ratio) showed increased brush use, suggesting that individual brush use increases with increasing availability. However, questions remain as to the effects of larger groups given that some commercial farms employ much larger group sizes than those tested in this study (e.g., 100–300 cows/pen; [15]). Further work is required to understand changes in brush use in these conditions; however, cautious extrapolation of our results would indicate that average brush use/day/cow would be very low in these large groups sharing a single brush. 

We found higher brush use when the brush was placed close to the feed and water resources, as did previous work [9]. Although our study did not set out to formally record contamination of feed or water with cow hair or dust, we noted no evidence of contamination despite increased use when close to these resources. Placing the brush closer to feed and water may increase overall brush use, although the close proximity of multiple high valued resources may limit access for some cows (e.g., subordinates) when competition for any of these resources is high. Brush use did not differ between daytime and nighttime in any of the group sizes tested in our study, and the brush was always used for less than 50% of the 24 h. However, brush use still increased when the group size and cow-to-brush ratio were lower. Lactating cows may have a limited time budget available for using the brush. Given that many behaviors are synchronized in groups of cows [16], social factors may limit brush use by some individuals when stocking density is high. In our study cows were habituated to the entire pen area and brush location before the data collection and it is reasonable to assume that cows knew where the brush was located. However, despite a high motivation for brushing [5], the non-central location of the brush in the larger groups used in this study may have reduced cows’ brush exposure in association with spatial preferences. Going to the other end of the pen to brush may have been too costly for some cows, thus reducing the overall use as group and pen size increased. We suggest that future work investigates how brush use duration relates to time budgets of individual cows (i.e., accounting for time spent away from the pen, feeding, or lying).

Previous work found a decrease in the frequency of brush use events with increasing heat load [9], which may suggest a relationship between ambient temperature and brush use. The second replicate of our study corresponds to a period with low ambient temperatures with daily averages sometimes below −10 °C. Although this was unlikely to induce cold stress in mature lactating cows [17,18], floor surfaces were slipperier in the brush area and fly load was likely to be low [19] likely contributing to reduced brush use. 

Individual variation between the cows enrolled in the two replicates may also explain some of the differences observed. Although we did not distinguish between individual cows in this study, previous findings indicated considerable between-individual variation in the daily brush use [8,14]. Grooming is arguably less urgent than other behaviors such as feeding or resting and thus may be one of the first to decline in the days before clinical illness is observed [20]. Previous work found that cattle experiencing metritis [12], lameness [21], and social stress [22,23], all showed evidence of reduced brush use. To facilitate the practical use of brushing behavior as an early illness indicator, future work should investigate whether healthy individuals vary in terms of brush use based on placement within the pen and whether competition for other resources affects brush use of individuals. 

We altered the group size and the cow-to-brush ratio simultaneously and so are unable to determine which factor contributes to the differences in brush use across treatments. Cow behavior and resource use are known to be affected by both group size (e.g., [24]) and stocking density (e.g., [25]). We encourage future research to employ automated brushes to disentangle the effects of group size and cow-to-brush ratio.

## 5. Conclusions

Using an automated brush, we found that cows used the brush for longer when it was placed close to feed and water. Average brush use per cow was also higher in smaller groups with a lower cow-to-brush ratio. We suggest that future research assesses the use multiple automated brushes in each pen.

## Figures and Tables

**Figure 1 animals-13-00760-f001:**
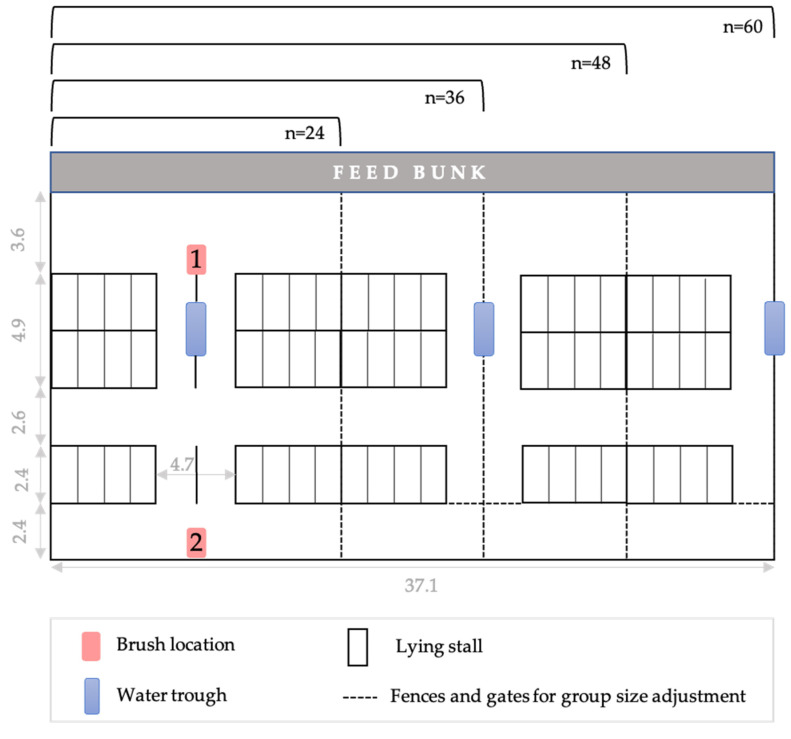
Pen layout for different group sizes (n = 24, 36, 48, 60) of lactating cows, showing the two locations (1: feed alley, 2: back alley) where the automated cylindrical-shaped brush was provided. Pen dimensions are shown in m.

**Figure 2 animals-13-00760-f002:**
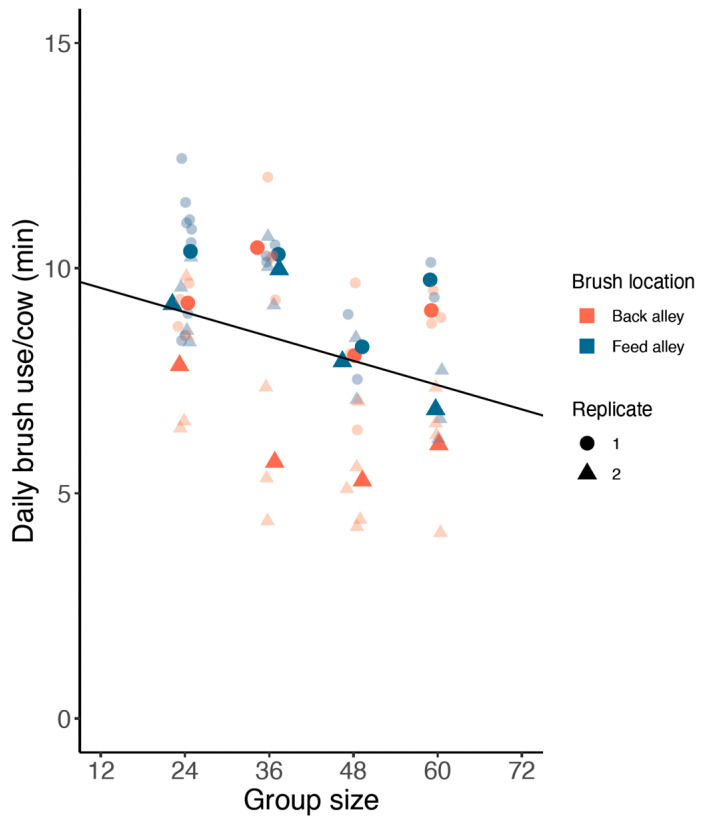
Solid colored points show the average daily brush use per cow across days based on a single mechanical brush placed in four different group size conditions positioned in the feed alley or in the back alley; data are reported separately for each of two replicates. Small transparent points show the average brush use values per cow for the days analyzed in each treatment condition. Points are spread out around respective group size on the x axis to avoid overlap. Fitted line indicates the linear relationship between group size and average daily brush use duration per cow.

## Data Availability

Data and analysis code are available at https://doi.org/10.5683/SP3/YJU5ML.

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
