# Peer review of "The Effect of Placement and Group Size on the Use of an Automated Brush by Groups of Lactating Dairy Cattle"

_animals, 2023, doi:10.3390/ani13040760_

Round 1
Reviewer 1 Report
I enjoyed reading your article, research results and the way you present them. Nevertheless, I have some comments regarding the content of the article.
Regardless of the general formulation of the purpose of the research / study, it would be worth writing, in my opinion, what was the cognitive (scientific) purpose of the research undertaken, and what was the utilitarian (useful) purpose of the research. In my opinion, it would be worth summarizing the review of the state of knowledge in the Introduction with the formulation of the research problem (The research problem is …). Of course, all important information about the research problem has been presented in the Introduction, but it would be worth summarizing this review of the state of knowledge with the statement: "The research problem is ...". The research problem can be associated with the indication of a gap in the current state of knowledge, which was the premise for undertaking a research study. General information related to the research gap has been given, it only needs to be called a research gap.
Could placing the brush near the water and feed areas increase the risk of contamination with dust or other particles as a result of brushing the cows' hair? Maybe it's worth mentioning in the discussion of research results?
In the Materials and Methods chapter, I am missing detailed information about the brush used in the experiment. It is true that the name of the manufacturer of this device and the equipment of the brush allowing for automatic data collection are given. However, in my opinion, the positioning of the rotating brush in relation to the walking/manure (slurry) alley is also important. I guess that the rotating brush was rigidly mounted, without the possibility of changing the direction of the axis of rotation. If the axis of rotation of the brush was located transversely to the direction of movement of the scrapers in the manure alley, then the cow stood in such a place during brushing that it largely covered the passage (for other cows) between the two manure corridors. If the axis of rotation of the brush was located longitudinally to the direction of movement of the scrapers in the manure alley, then the cow stood during brushing so that the front and rear legs should stand at different levels. Between the manure corridor and the passage between the corridors there is a threshold with a certain height (about 10 cm or more?). Without this threshold, it would not be possible for the manure scraper to work without losses. Such my considerations result from the analysis of details in the form of symbols shown in Figure 1.
What was the shape (and diameter) of the brush used in the experiment? Of course, such information can be found in available sources based on the name of the brush model and its manufacturer. In my opinion, it would be worth specifying in the description of the experiment whether the brush was cylindrical or hourglass-shaped. In addition, it would be useful to provide more details on the operation of the brush, including its rotational speed, the time after which the brush automatically switches off after the cow has left. The latter information is important for the accuracy of collecting data on cow brushing times. In lines 73-74 there is the sentence "Rotation start times and durations were recorded continuously ...". In some models of brushes, they automatically turn off a few seconds after brushing the cow. Has this detail been considered in the research data collection process? The question can be formulated in another way: Is the time of effective brushing of cows and the time of work (rotating) of the brush really the same? This is due to the technical and operational characteristics of the brush, including its adjustment, and it is worth mentioning in the article.
You can write whether there were any technical problems with the brush during the experiment that required intervention (repairs, adjustments, etc.). At what height was the brush placed above the floor surface in the barn where the cows walked? On what basis was this height determined? How was the height of the brush above the floor related to the height of the cows? What was the average height of the cows included in the experiment and what was the standard deviation of this height? What was the maximum height difference between the cows in the experiment. Could this difference have influenced the results of the study? Of course, this difference in height between cows can be compensated by equipping the brush mounting system with appropriate shock absorbers to reduce the risk of brushing very tall cows too aggressively. It is worth writing about such details as shock absorbers (if used) and others in the Materials and Methods chapter. Thanks to this, the description of the research material will be more accurate and will allow for a more accurate interpretation of the research results.
The brush is placed in the alley where the manure (slurry) scraper works. Doesn't this placement of the brush conflict a bit with the manure scraper? I mean the situation where the cow is being brushed and at the same time the manure scraper moves under her legs. Could this cause the cow to stop brushing if the manure scraper runs over her legs? I think it is worth answering these questions in the context of planning future research on the use of the brush in connection with its location and layout of the most important parts of the pen.
I have comments on Figure 1. In the legend, the brush is marked with a circle, and in the figure - with an oval. But maybe it doesn't matter that much because the colour is important. I suggest that in the Figure, instead of the oval shape of the figure representing the brush, draw a rectangle or an hourglass (corresponding to the actual shape of the brush in the experiment). It is important that the position of the rectangle or hourglass in the Figure results from the position of the axis of rotation of the brush. There are modular pens in the barn with 12 lying stalls each (according to the content of the article), so in my opinion there are two short lines (continuous lines) missing on the axis connecting brush 1 and brush 2. One of these short lines would be on the length of two stalls (head to head) and the second short line along the length of the lying stall in the bottom row. In my opinion, these are permanent fences, so you need to draw them with a solid line. I also note the center dotted line drawn vertically (passing through the water trough). In my opinion, this line cannot be a dotted line all the way through. Moving gates are only located above the manure corridors. In the remaining part, these are permanent (and not movable) fences that need to be drawn with a continuous line.
It is worth mentioning in the article that the lower, horizontal manure corridor (with brush 2) with standard cow maintenance functions as a passage corridor for moving cows to the milking parlour. This corridor was adapted (I presume) for the brush experiment.
I am wondering about the given issue of brushes working for no more than 50% during the day. Maybe the working time of the brushes should be related to the effective time the cows stay in the pen, i.e. excluding the time when the cows are in the parlour? This is only a proposal. If the herd of cows was not recorded during the experiment, it will probably be difficult to determine exactly how much time the cows spent in the parlour during the day.
The material and research results presented in the article inspire to develop further research, including those related to the structure of time spent by cows on various activities in the pen and outside the pen (milking). Therefore, I trust that the subject will be developed and will bring many more interesting observations, research results and publications.
Author Response
From reviewer 1
Comments and Suggestions for Authors
I enjoyed reading your article, research results and the way you present them. Nevertheless, I have some comments regarding the content of the article.
AU: Thank you for your comments!
Regardless of the general formulation of the purpose of the research / study, it would be worth writing, in my opinion, what was the cognitive (scientific) purpose of the research undertaken, and what was the utilitarian (useful) purpose of the research. In my opinion, it would be worth summarizing the review of the state of knowledge in the Introduction with the formulation of the research problem (The research problem is …). Of course, all important information about the research problem has been presented in the Introduction, but it would be worth summarizing this review of the state of knowledge with the statement: "The research problem is ...". The research problem can be associated with the indication of a gap in the current state of knowledge, which was the premise for undertaking a research study. General information related to the research gap has been given, it only needs to be called a research gap.
AU: Thank you for this suggestion. We have now summarized the research problem in the Introduction L59-62 as follows:
“In summary, the research problem is twofold: there is a lack of scientific knowledge about how cows use mechanical brushes in groups of different size and with varying location, and there are no practical devices available to automatically monitor brushing behavior in dairy cow groups. To address this research gap, we aimed to establish an automated brush prototype and investigate how the use of a single mechanical brush is affected by (1) increasing group sizes and (2) placement of the brush in relation to proximity to feed and water. We hypothesized that cows would use the brush more when housed in smaller groups and when the brush was placed closer to the feed and water.”
Could placing the brush near the water and feed areas increase the risk of contamination with dust or other particles as a result of brushing the cows' hair? Maybe it's worth mentioning in the discussion of research results?
AU: We did not observe any contamination, we added this to L189-191:
“Although our study did not set out to formally record contamination of feed or water with cow hair or dust, we noted no evidence of contamination despite increased use when close to these resources.”
In the Materials and Methods chapter, I am missing detailed information about the brush used in the experiment. It is true that the name of the manufacturer of this device and the equipment of the brush allowing for automatic data collection are given. However, in my opinion, the positioning of the rotating brush in relation to the walking/manure (slurry) alley is also important. I guess that the rotating brush was rigidly mounted, without the possibility of changing the direction of the axis of rotation. If the axis of rotation of the brush was located transversely to the direction of movement of the scrapers in the manure alley, then the cow stood in such a place during brushing that it largely covered the passage (for other cows) between the two manure corridors. If the axis of rotation of the brush was located longitudinally to the direction of movement of the scrapers in the manure alley, then the cow stood during brushing so that the front and rear legs should stand at different levels. Between the manure corridor and the passage between the corridors there is a threshold with a certain height (about 10 cm or more?). Without this threshold, it would not be possible for the manure scraper to work without losses. Such my considerations result from the analysis of details in the form of symbols shown in Figure 1.
What was the shape (and diameter) of the brush used in the experiment? Of course, such information can be found in available sources based on the name of the brush model and its manufacturer. In my opinion, it would be worth specifying in the description of the experiment whether the brush was cylindrical or hourglass-shaped. In addition, it would be useful to provide more details on the operation of the brush, including its rotational speed, the time after which the brush automatically switches off after the cow has left. The latter information is important for the accuracy of collecting data on cow brushing times. In lines 73-74 there is the sentence "Rotation start times and durations were recorded continuously ...". In some models of brushes, they automatically turn off a few seconds after brushing the cow. Has this detail been considered in the research data collection process? The question can be formulated in another way: Is the time of effective brushing of cows and the time of work (rotating) of the brush really the same? This is due to the technical and operational characteristics of the brush, including its adjustment, and it is worth mentioning in the article.
AU: Thank you for raising these important points. We now added more details to the Materials and Methods section to answer these questions, see L78-98:
“One cylindrical-shaped mechanical brush (Luna, Lely, Maassluis, The Netherlands) was placed in a pen designed to house 60 cows and adapted to this study by the inclusion of an additional alley (Figure 1). The brush was mounted 0.4 m above the shoulder height of the shortest cow according to manufacturer instructions. At both locations tested the brush was placed such that the rotation axis was transverse to the movement direction of manure scrapers, allowing cows to step over the moving scraper while brushing. The brush was fitted with an electronic interface and two light sensors to automatically detect when the swinging brush arm is deflected from the vertical position and the brush rotates. Brush rotation was 30 rpm in the direction opposite to that of the push and stopped if the brush remained in the vertical position for 4 s. Rotation direction, start times and durations were recorded continuously on a MicroSD card in the electronic interface attached to the brush and saved daily on a computer while cows were away from the pen during milking. In a preliminary validation, we found that brush use events in the data (i.e., a new event corresponding to each change in rotation direction and rotation re-start) did not necessarily reflect separate brush use bouts of cows. Preliminary observations indicated that cows often stop and resume brushing, or multiple cows take turns in using the brush, indicating that brush rotation frequency may be influenced by the general barn environment and its daily variation (e.g., animal and human movement). However, the total daily brush use duration by the group of cows was well represented by the rotation data. Therefore, instead of brush use frequency, we calculated total daily rotation time to reflect the daily brush use duration by the group.”
You can write whether there were any technical problems with the brush during the experiment that required intervention (repairs, adjustments, etc.). At what height was the brush placed above the floor surface in the barn where the cows walked? On what basis was this height determined? How was the height of the brush above the floor related to the height of the cows? What was the average height of the cows included in the experiment and what was the standard deviation of this height? What was the maximum height difference between the cows in the experiment. Could this difference have influenced the results of the study? Of course, this difference in height between cows can be compensated by equipping the brush mounting system with appropriate shock absorbers to reduce the risk of brushing very tall cows too aggressively. It is worth writing about such details as shock absorbers (if used) and others in the Materials and Methods chapter. Thanks to this, the description of the research material will be more accurate and will allow for a more accurate interpretation of the research results.
AU: We encountered days with missing data due to technical errors when writing to the SD card; these days were excluded from the analysis. The brush was mounted following manufacturer’s instructions based on the measurement of the shortest cow. To obtain this, we only measured a few short cows in the trial and did not measure the height of all cows in the herd but preliminary tests of the brush in a dynamic group of cows over 2 months indicated no problems with too short or too tall animals. The brush is mounted on a swinging arm and did not include any shock absorbers. We now added these details to the manuscript L131-134:
“We excluded days with missing data due to technical errors with writing to the microSD card in the brush interface and analyzed group-level brush use data for a minimum of two days (mean ± SD: 3.7 ± 1.9 days) for each group size and brush placement treatment using R 4.2.0 (R Core Team, Vienna, Austria).”
The brush is placed in the alley where the manure (slurry) scraper works. Doesn't this placement of the brush conflict a bit with the manure scraper? I mean the situation where the cow is being brushed and at the same time the manure scraper moves under her legs. Could this cause the cow to stop brushing if the manure scraper runs over her legs? I think it is worth answering these questions in the context of planning future research on the use of the brush in connection with its location and layout of the most important parts of the pen.
AU: Thank you for this comment. We did not detect disturbances due to the manure scraper, cows were well habituated to this equipment and appeared to easily step over the scraper while brushing. Placing the brush in the alleys allowed the surface to be regularly cleaned, likely leading to reduced slipperiness of the standing area standing while being brushed. We now included this information in L81-83 (see above).
I have comments on Figure 1. In the legend, the brush is marked with a circle, and in the figure - with an oval. But maybe it doesn't matter that much because the colour is important. I suggest that in the Figure, instead of the oval shape of the figure representing the brush, draw a rectangle or an hourglass (corresponding to the actual shape of the brush in the experiment). It is important that the position of the rectangle or hourglass in the Figure results from the position of the axis of rotation of the brush. There are modular pens in the barn with 12 lying stalls each (according to the content of the article), so in my opinion there are two short lines (continuous lines) missing on the axis connecting brush 1 and brush 2. One of these short lines would be on the length of two stalls (head to head) and the second short line along the length of the lying stall in the bottom row. In my opinion, these are permanent fences, so you need to draw them with a solid line. I also note the center dotted line drawn vertically (passing through the water trough). In my opinion, this line cannot be a dotted line all the way through. Moving gates are only located above the manure corridors. In the remaining part, these are permanent (and not movable) fences that need to be drawn with a continuous line.
AU: Thank you for pointing this out. We adjusted Figure 1 to better reflect the brush shape and barn layout.
It is worth mentioning in the article that the lower, horizontal manure corridor (with brush 2) with standard cow maintenance functions as a passage corridor for moving cows to the milking parlour. This corridor was adapted (I presume) for the brush experiment.
AU: We now added this to the Materials and Methods at L79-80 (see above).
I am wondering about the given issue of brushes working for no more than 50% during the day. Maybe the working time of the brushes should be related to the effective time the cows stay in the pen, i.e. excluding the time when the cows are in the parlour? This is only a proposal. If the herd of cows was not recorded during the experiment, it will probably be difficult to determine exactly how much time the cows spent in the parlour during the day.
AU: Thanks for this suggestion. Indeed, cows were brought to the parlour for milking and came back on their own as they have finished, making it difficult to know how much time each cow spent away from the pen. However, we expect this time to be similar across the treatments we applied and given the relatively short duration of milking we don’t expect that subtracting these times would considerably change the percentage of time the brush was not used. We now added that future research could investigate brush use in relation to each cow’s available time budget, including time spent away from the pen, see L205-207:
“We suggest that future work investigates how brush use duration relates to time budgets of individual cows (i.e., accounting for time spent away from the pen, feeding, or lying).”
The material and research results presented in the article inspire to develop further research, including those related to the structure of time spent by cows on various activities in the pen and outside the pen (milking). Therefore, I trust that the subject will be developed and will bring many more interesting observations, research results and publications.
AU: Thank you! We agree that this short communication is an important first step to develop a thorough understanding of the factors that influence brush use through automated recording methods.
Reviewer 2 Report
This is an interesting, well-designed and described paper by Foris et al that questions how the placement of automated brushes as well as group size impact brush use. The the study design is straight-forward and the model findings clear as cows favor the use of the brush when it is near to feed and water as well as use the brush more when housed in smaller groups. The implication for commercial herds with large numbers of cows in a pen is that multiple brushes would increase brush use and presumably cow welfare.
The data presented appears to be a little more complicated than the model results as the average daily brush use/cow in largest group size in the first replicate is the same or greater than the smallest group. Furthermore, it appears that most of the effect between front and back alleyways occurs in the second replicate (especially for intermediate size groups and are there any interaction between location, group size and replicate?). Perhaps the authors could consider an alternative data presentation where the individual daily brush use per cow data points are shown with the means as the models are seeing the full data set and not just the means. Having a better sense of the range of variation in the data might make the outcomes more convincing to the reader.
One other minor suggestions would be in the graph legend for Figure 2 where back versus front alley is denoted would be to either use a neutral shape (large square) or include both circle and triangle. The reader can be confused that the front and back alleyway only pertains to the first replicate as it is denoted with a circle.
Also the authors don't mention in the methods if standard quality control checks for linear models including homoscedasticity or normality of the residuals etc were performed.
Author Response
Reviewer 2
This is an interesting, well-designed and described paper by Foris et al that questions how the placement of automated brushes as well as group size impact brush use. The the study design is straight-forward and the model findings clear as cows favor the use of the brush when it is near to feed and water as well as use the brush more when housed in smaller groups. The implication for commercial herds with large numbers of cows in a pen is that multiple brushes would increase brush use and presumably cow welfare.
The data presented appears to be a little more complicated than the model results as the average daily brush use/cow in largest group size in the first replicate is the same or greater than the smallest group. Furthermore, it appears that most of the effect between front and back alleyways occurs in the second replicate (especially for intermediate size groups and are there any interaction between location, group size and replicate?). Perhaps the authors could consider an alternative data presentation where the individual daily brush use per cow data points are shown with the means as the models are seeing the full data set and not just the means. Having a better sense of the range of variation in the data might make the outcomes more convincing to the reader.
One other minor suggestions would be in the graph legend for Figure 2 where back versus front alley is denoted would be to either use a neutral shape (large square) or include both circle and triangle. The reader can be confused that the front and back alleyway only pertains to the first replicate as it is denoted with a circle.
AU: Thank you for these suggestions, we now modified Figure 2 to show the daily brush use per cow values with transparent colors in addition to the means and adjusted the legend for brush location to avoid confusion. We have now added that no interaction effects were found (L137-138):
“We did not detect any significant interactions so these terms were excluded from the final model.”
Also the authors don't mention in the methods if standard quality control checks for linear models including homoscedasticity or normality of the residuals etc were performed.
AU: We now added this information at L144-145:
“Model residuals were plotted and inspected visually to confirm homoscedasticity and normality.”
Reviewer 3 Report
In my opinion, the communication that authors presented could be used as a poster communication in a conference or something similar, or even the start or a bigger study. However, I feel the data and results are not enough for a Q1 journal.
If you search in pubmed "brush dairy cattle" it just appeared 69 results about this topic, with studies that show differences due to jierarchy, time to use, density, etc.
I encourage authors to re-think the experimental design.
Author Response
Reviewer 3
Comments and Suggestions for Authors
In my opinion, the communication that authors presented could be used as a poster communication in a conference or something similar, or even the start or a bigger study. However, I feel the data and results are not enough for a Q1 journal.
If you search in pubmed "brush dairy cattle" it just appeared 69 results about this topic, with studies that show differences due to jierarchy, time to use, density, etc.
I encourage authors to re-think the experimental design.
AU: This study – intended to be published as short communication, not full research paper – aimed to build upon what is known from previous work (e.g., the above-mentioned studies) and 1) establish a prototype to automatically collect brush use information and 2) investigate some of the potential main factors influencing brush use. We agree that there is certainly more work to be done on this topic, but believe that sharing these results with the research community can facilitate the automated measurement of brush use with simple technology and may lead to improved guidance for farmers on how to provide brushes for their cows.
Round 2
Reviewer 3 Report
Authors just answered my query, but they did not change anything at all, nor tried to justify properly their study. Therefore, I still thinking that the novelty is very low and there are some studies already published in this area. This one does not increase the knowledge in the area, nor include new parameters or different data.
I encourage authors to re-think the experimental design.